# RPGAN: RANDOM PATHS AS A LATENT SPACE FOR GAN INTERPRETABILITY

## ABSTRACT

In this paper, we introduce Random Path Generative Adversarial Network (RP-GAN) — an alternative design of GANs that can serve as a tool for generative model analysis. While the latent space of a typical GAN consists of input vectors, randomly sampled from the standard Gaussian distribution, the latent space of RPGAN consists of random paths in a generator network. As we show, this design allows to understand factors of variation, captured by different generator layers, providing their natural interpretability. With experiments on standard benchmarks, we demonstrate that RPGAN reveals several interesting insights about the roles that different layers play in the image generation process. Aside from interpretability, the RPGAN model also provides competitive generation quality and allows efficient incremental learning on new data.

## 1 INTRODUCTION

Nowadays, deep generative models are an active research direction in the machine learning community. The dominant methods for generative modeling, such as Generative Adversarial Networks (GANs), are currently able to produce diverse photorealistic images (Brock et al., 2019; Karras et al., 2019). These methods are not only popular among academicians, but are also a crucial component in a wide range of applications, including image editing (Isola et al., 2017; Zhu et al., 2017), super-resolution (Ledig et al., 2017), video generation (Wang et al., 2018) and many others.

Along with practical importance, a key benefit of accurate generative models is a more complete understanding of the internal structure of the data. Insights about the data generation process can result both in the development of new machine learning techniques as well as advances in industrial applications. However, most state-of-the-art generative models employ deep multi-layer architectures, which are difficult to interpret or explain. While many works investigate interpretability of discriminative models (Zeiler & Fergus, 2014; Simonyan et al., 2013; Mahendran & Vedaldi, 2015), only a few (Chen et al., 2016; Bau et al., 2019) address the understanding of generative ones.

In this work, we propose the Random Path GAN (RPGAN) — an alternative design of GANs that allows natural interpretability of the generator network. In traditional GAN generators, the stochastic component that influences individual samples is a noisy input vector, typically sampled from the standard Gaussian distribution. In contrast, RPGAN generators instead use stochastic routing during the forward pass as their source of stochasticity. In a nutshell, the RPGAN generator contains several instances of the corresponding layer. For each sample, only one random instance of each layer is activated during generation. The training of the RPGAN can then be performed in the same adversarial manner as in traditional GANs. In the sections below, we show how RPGAN allows to understand the factors of variation captured by the particular layer and reveals several interesting findings about the image generation process, e.g. that different layers are "responsible for" coloring or objection location. As a practical advantage, RPGANs can be efficiently updated to new data via the simple addition of new instances to the bucket, avoiding re-training the full model from scratch. Finally, we observe that RPGANs allow the construction of generative models without nonlinearities, which can significantly speed up the generation process for fully-connected layers.

In summary, the main contributions of our paper are the following:

- We introduce RPGAN — GAN with an alternative source of stochasticity, based on random routing. While being close to traditional GANs in terms of generation quality, RPGAN allows natural interpretability and efficient model updates with new data.

- With extensive experiments on standard benchmarks we reveal several insights about the image generation process. Many of our insights confirm and extend recent findings from Bau et al. (2019). Note, that our scheme is more general compared to the technique from Bau et al. (2019) as RPGAN does not require labeled datasets or pretrained segmentation models.

- We open-source the PyTorch implementation of RPGAN with common generator architectures[1].

The rest of this paper is organized as follows. In Section 2 we review relevant ideas from prior art. The proposed Random Path GAN design is described in Section 3 and experimentally evaluated in Section 4. Section 5 concludes the paper and discusses possible directions for future work.

## 2   RELATED WORK

In this section we briefly describe connections of RPGAN to existing ideas from prior works

**Generative adversarial networks.** GANs are currently one of the main paradigms in generative modelling. Since the seminal paper on GANs by Goodfellow et al. (2014), a plethora of alternative loss functions, architectures, normalizations, and regularization techniques were developed (Kurach et al., 2019). Today, state-of-the-art GANs are able to produce high-fidelity images, often indistinguishable from real ones (Brock et al., 2019; Karras et al., 2019). In essence, GANs consist of two networks – a generator and a discriminator, which are trained jointly in an adversarial manner. In standard GANs, the generation stochasticity is provided by the input noise vector. In RPGANs, we propose an alternative source of stochasticity by using a fixed input but random routes during forward pass in the generator.

**Specific GAN architectures.** Many prior works investigated different design choices for GANs, but to the best of our knowledge, none of them explicitly aimed to propose an interpretable GAN model. Hoang et al. (2018) proposed the use of several independent generators to address the mode collapse problem. Chavdarova & Fleuret (2018) employ several auxiliary local generators and discriminators to improve mode coverage as well. Huang et al. (2017) use layer-wise generators and discriminators to enforce hidden representations produced by layers of the generator to be similar to the corresponding representations produced by a reversed classification network. Important differences of RPGAN compared to the works described above is that it uses random routes as its latent space and does not enforce to mimic the latent representations of pretrained classifiers.

**Interpretability.** While interpretability of models based on deep neural networks is an important research direction, most existing work addresses the interpretability of discriminative models. These works typically aim to understand the internal representations of networks (Zeiler & Fergus, 2014; Simonyan et al., 2013; Mahendran & Vedaldi, 2015; Dosovitskiy & Brox, 2016) or explain decisions produced by the network for particular samples (Sundararajan et al., 2017; Bach et al., 2015; Simonyan et al., 2013). However, only a few works address interpretability of generative models. A related work by Bau et al. (2019) develops a technique that allows to identify which parts of the generator are responsible for the generation of different objects. In contrast, we propose GANs with alternative source of stochasticity that allows natural interpretation by design. Some of our findings confirm the results from Bau et al. (2019), which provides stronger evidence about the responsibilities of different layers in the generation process. Note, that the technique (Bau et al., 2019) requires a pretrained segmentation network and cannot be directly applied to several benchmarks, e.g. CIFAR-10 or MNIST. In contrast, RPGAN does not require any auxiliary models or supervision and can be applied to any data.

---

[1]https://github.com/rpgan-ICLR2020/RPGAN

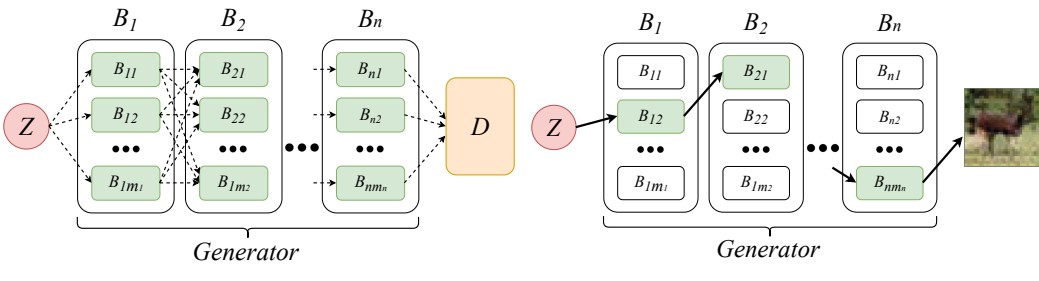

(a) The generator consists of buckets, each containing several blocks.

(b) Forward pass.

Figure 1: The RPGAN generator. During forward pass, only one block in each bucket is activated.

# 3 RANDOM PATH GAN

## 3.1 MOTIVATION

Before the formal description, we provide an intuition behind the RPGAN model. Several prior works have demonstrated that in discriminative convolutional neural networks different layers are "responsible" for different levels of abstraction (Zeiler & Fergus, 2014; Babenko et al., 2014). For instance, earlier layers aim to detect small texture patterns, while activations in deeper layers typically correspond to semantically meaningful concepts. Similarly, in our paper we aim to understand the roles that different GAN layers play in image generation. Thus, we propose an architecture that provides a direct way to interpret the impact of individual layers. For a given generator architecture, we construct several copies of each layer in its architecture. During the forward pass we randomly choose a layer instance that will be used when generating a particular image. Therefore, we can analyze the role of each RPGAN layer by visualizing how different instances of that layer affect the generated image.

## 3.2 MODEL

Here we formally describe the structure of the RPGAN model. The model is highly flexible to the choice of generator and discriminator architectures as well as to the loss function and learning strategy. Similarly to the standard GAN architectures, our model consists of two networks – a generator and a discriminator. The RPGAN discriminator operates exactly like discriminators in common GANs, hence below we focus on the generator description.

Unlike existing GANs, the RPGAN generator always receives a fixed input vector $Z$ during forward pass and aims to produce an image from the real image distribution. The generator consists of several consequent *buckets* $B_1, \ldots, B_n$. Each bucket is a union of independent *blocks*: $B_i = \{B_{i1}, \ldots, B_{im_i}\}$, where each block is an arbitrary computational unit and $m_i = |B_i|$. A typical example of a block is a ResNet block (He et al., 2015), a convolutional layer with a nonlinearity or any other (see Figure 1a) layer type. In our experiments, all the units from the same bucket have the same architecture.

For each $i = 1, \ldots, n-1$ a block from the bucket $B_i$ produces an intermediate output tensor that is passed to a block from the next bucket $B_{i+1}$. Typically we associate each bucket with a layer (or several layers) in the generator architecture, which we aim to interpret or analyze. A block from the first bucket $B_1$ always receives a fixed input vector $Z$, which is the same for different forward passes. The stochasticity of the generator arises from a random path that goes from $Z$ to an output image, using only a single block from each bucket. Formally, during each forward pass, we randomly choose indices $s_1, \ldots, s_n$ with $1 \leq s_i \leq m_i$. The generator output is then computed as $B_{ns_n} \circ \cdots B_{2s_2} \circ B_{1s_1}(Z)$ (see Figure 1b). Thus, the generator defines a map from the Cartesian product $\langle m_1 \rangle \times \langle m_2 \rangle \times \cdots \times \langle m_n \rangle$ to the image space. Note that we can take an arbitrary existing GAN model, group its generator layers into buckets and replicate them into multiple blocks. In these terms, the original model can be treated as the RPGAN model with a single block in each bucket and

random input noise. Note that during image generation we perform the same number of operations as in the standard GAN generator.

By its design, RPGAN with buckets $B_1, \ldots, B_n$ and a constant input $Z$ is able to generate at most $|B_1| \times \cdots \times |B_n|$ different samples were $|B_k|$ is the number of blocks in the bucket $B_k$. Nevertheless, this number is typically much larger compared to the training set size. We argue that the probability space of random paths can serve as a latent space to generate high-quality images, as confirmed by the experiments below.

**Block diversity loss.** To guarantee that blocks in a particular bucket are different, we also add a specific diversity term in the generator loss function. The motivation for this term is to prevent blocks $B_{ki}, B_{kj}$ from learning the same weights. Let $W$ be the set of all parameters of the generator. For each parameter $w \in W$ there is a set of its instances $\{w^{(1)}, \ldots w^{(m_w)}\}$ in the RPGAN model. Then we enforce the instances to be different by the loss term

$$-\sum_{w \in W, i \neq j} \text{MSE}\left(\frac{w^{(i)}}{s_w}, \frac{w^{(j)}}{s_w}\right)$$

Here we also normalize by the standard deviation $s_w$ of all parameters from different blocks that correspond to the same layer. This normalization effectively guarantees that all buckets contribute to the diversity term.

## 4 Experiments

**Architecture.** In all the experiments in this section, we use ResNet-like generators with spectral normalization and the hinge loss (SN-ResNet) described in Miyato et al. (2018). The blocks in the first bucket are fully-connected layers, the blocks in the last bucket are convolutional layers and blocks in all other buckets are residual blocks with two convolutions and a skip connection. If not stated otherwise, all the buckets have the same number of blocks. Additional experiments with other architectures are provided in Appendix.

**Datasets.** We performed experiments on CIFAR-10 (Krizhevsky et al., 2009), LSUN-bedroom (Yu et al., 2015) and Anime Faces (Jin et al., 2017) datasets. For different datasets we use different numbers of discriminator steps per one generator step $d_{steps}$ and different numbers of blocks in a bucket $n_{blocks}$. We summarize the main parameters used for three datasets in Table 1. In the last column we also report *Coverage*, which is the ratio of the latent space cardinality (which equals the number of buckets to the power $n_{blocks}$) to the dataset size. Intuitively, large coverage guarantees that RPGAN has a sufficiently rich latent space of generator routes to capture the reference dataset. In the experiments below, we demonstrate that even moderate coverage is sufficient to generate high-fidelity images (see the LSUN-bedroom dataset with coverage $\approx 3.3$).

| Dataset | Image size | Number of buckets | $n_{blocks}$ | $d_{steps}$ | Batch size | Coverage |
|---------|-----------|-------------------|--------------|-------------|------------|----------|
| CIFAR-10 | $32 \times 32$ | 5 | 40 | 5 | 64 | 2048 |
| Anime Faces | $64 \times 64$ | 6 | 20 | 1 | 32 | $\approx 2970$ |
| LSUN-bedroom | $128 \times 128$ | 7 | 10 | 1 | 16 | $\approx 3.3$ |

Table 1: The details of architectures and training protocols used for different datasets.

**Training details.** We use the Adam optimizer with learning rate equal to $0.25 \times 10^{-3}$, $\beta_1, \beta_2$ equal to $0.5, 0.999$ and train the model for $45 \times 10^4$ generator steps for CIFAR-10 and $25 \times 10^4$ generator steps for Anime Faces and LSUN-bedroom datasets. During training we also learn the unique input vector $Z$. We observed that a learnable $Z$ slightly improves the final generation quality and stabilizes the learning process. During the training step, we pass $Z$ through $N$ independent random paths. Formally, let $\{x_1, \ldots, x_N\}$ be a batch of samples received from a bucket $B_k$. To pass this batch through the bucket $B_{k+1}$ we take random blocks $B_{ki_1}, \ldots, B_{ki_N}$ and form a new batch $\{B_{ki_1}(x_1), \ldots, B_{ki_N}(x_N)\}$. In all the experiments, we use the same training protocols for both the RPGAN and the standard GAN of the generator architecture. Note, that despite larger number of learnable parameters, RPGAN does not require more data or training time to achieve the same quality, compared to standard GANs.

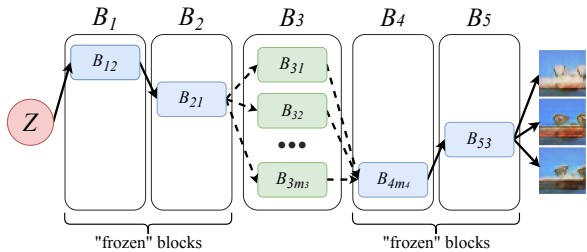

Figure 2: The five-bucket RPGAN model with "frozen" blocks in buckets 1, 2, 4 and 5. In this case the stochasticity comes only from the third bucket. Analyzing the generated images allows to identify factors of variation, influenced by the third bucket.

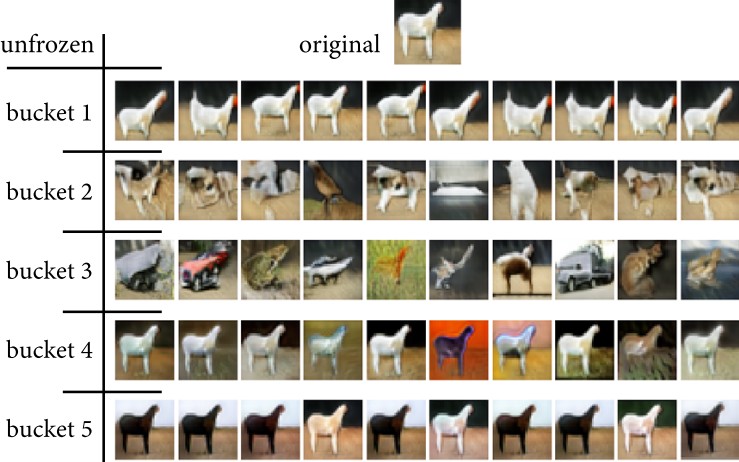

Figure 3: *Top image*: sample generated with a fixed sequence of blocks. *Horizontal lines*: samples generated by the same sequence of blocks in all buckets and one *unfrozen* bucket. In the selected bucket we choose ten arbitrary blocks to avoid excessively large figures. The samples produced in this way allow to interpret factors of variation, captured by different buckets.

### 4.1 LAYERS INTERPRETATION

In the first series of experiments we investigate the " responsibility areas" of different generator layers. This can be performed with a technique schematically presented on Figure 2. The goal is to interpret the role of the third bucket $B_3$ in a five-bucket generator. For all other buckets $B_1, B_2, B_4, B_5$ we fix arbitrary blocks, shown in blue on Figure 2. Then we generate images corresponding to routes that contain all the fixed blocks, with the stochasticity coming only from varying blocks inside the target bucket $B_3$. By inspecting the distribution of the obtained images, we can understand what factors of variation are influenced by $B_3$.

On Figure 3 we plot the samples generated with only one "unfrozen" bucket for one image from the CIFAR-10 dataset. Thus, each row shows how the original generated image could change if different blocks from the corresponding bucket are used. Several observations from Figure 3, Figure 9, Figure 12 and Figure 15 are listed below. The first bucket typically does not influence coloring and mostly affects small objects' deformations. The intermediate buckets have the largest influence on semantics. The last two buckets are mostly responsible for coloring and do not influence the content shape. In particular, on Figure 3 the fourth layer widely varies color, while the fifth acts as a general tone corrector. Note that these findings are consistent with the insights revealed by Bau et al. (2019).

To confirm the observations quantitatively, we perform the following experiment. We define a metric $d_{\mathbf{img}}$ that evaluates the similarity between two generated images. Different metrics are able to capture different variations (e.g. in terms of semantic, color histogram, etc.) and we describe two particular choices of $d_{\mathbf{img}}$ below. Then we choose a random route in the RPGAN generator and for each bucket

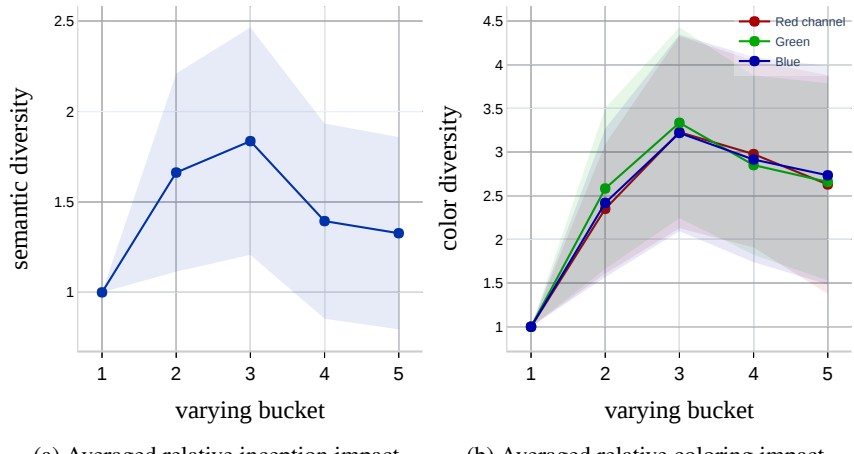

(a) Averaged relative inception impact    (b) Averaged relative coloring impact

Figure 4: Layer diversity compared to the first layer on CIFAR-10.

$B_l$ generate four images $s_1^{(l)}, \ldots, s_4^{(l)}$, varying blocks in $B_l$. In other words, rather then taking all possible pairs, we take four random samples from each line of the table in the Figure 3. Then we measure diversity w.r.t. $d_{\mathbf{img}}$ captured by bucket $B_l$ as a ratio

$$D_{l \to 1, d_{\mathbf{img}}} = \frac{\sum\limits_{i \neq j} d_{\mathbf{img}}(s_i^{(l)}, s_j^{(l)})}{\sum\limits_{i \neq j} d_{\mathbf{img}}(s_i^{(1)}, s_j^{(1)})}$$

Intuitively, we compute the relative diversity with respect to the first layer, which typically captures the smallest amount of variations in our experiments. We then average these ratios over 100 independent evaluations. High values of averaged ratio $D_{l \to 1, d_{\mathbf{img}}}$ imply higher diversity of the bucket $B_l$ compared to the first bucket in terms of the metric $d_{\mathbf{img}}$. For $d_{\mathbf{img}}$ we experimented with the following two metrics, capturing semantics and color differences correspondingly.

Inspired by the well-known Fréchet Inception Score concept (Heusel et al., 2017), we consider the Euclidean distance between the outputs of the last layer of the pretrained InceptionV3 network (Szegedy et al., 2016) for semantic distance evaluation. Namely, we define $d_{\mathbf{semantic}}(img_1, img_2)$ as $\|\mathbf{Iv3}(img_1) - \mathbf{Iv3}(img_2)\|_2$ were $\mathbf{Iv3}(img)$ is the InceptionV3 model activations for image $img$.

To measure differences in color, we take the Hellinger distance between the color histograms of generated samples. Namely, for each color channel, we split the range $[0, \ldots, 255]$ into 25 equal segments and evaluate the discrete distribution defined by the frequencies the sample's pixel intensities appear in a given segment. Then the Hellinger distance between two quantified color distributions is defined as $\frac{1}{\sqrt{2}}\sqrt{\sum\limits_{i=1}^{25}(\sqrt{p_i} - \sqrt{q_i})^2}$ . We compute this metric for each color channel independently.

The average values of $D_{l \to 1, d_{\mathbf{img}}}$ with the standard deviations are shown on Figure 4. It demonstrates that the semantic diversity is the largest for the intermediate layers. On the contrary, the last buckets, which are closer to the output, do not influence semantics but have higher impact in terms of color. Note that the first layer always shows the smallest variability in terms of both semantics and colors. The last bucket seems to be responsible for color correction and color inversion and has a lower pallet variability impact. Note, that the plots from Figure 4 reaffirm the findings coming from the Figure 3. Note that the similar empirical results also hold for other datasets (see figures 13, 16 in appendix).

Overall, we summarize the main findings common for CIFAR-10, LSUN and Anime Faces datasets as:

- The earlier layers have a smaller variability and seem to be responsible for the viewpoint and the position of the object on the image.

- The semantic details of the image content are mostly determined by the intermediate layers.
- The last layers typically affect only coloring scheme and do not affect content semantics or image geometry.

Note, that these conclusions can differ for other datasets or other generator architectures. For instance, for the four-bucket generator and MNIST (Figure 6, left) or randomly colored MNIST ( Figure 8, left) the semantics are mostly determined by the first two buckets.

## 4.2 RPGAN vs standard GAN

In this subsection, we argue that the interpretations of different layers, obtained with RPGAN, are also valid for the standard GAN generator of the same architecture. First, we demonstrate that both standard GAN and RPGAN trained under the same training protocol provide almost the same generation quality. As a standard evaluation measure, we use the Fréchet Inception Distance (FID) introduced in Heusel et al. (2017). We also compute precision-recall curve as defined in Sajjadi et al. (2018). For evaluation on CIFAR-10, we use 50000 generated samples and the whole train dataset. We also take ten independently trained generators and report minimal and average FID values. See Table 2 for FID comparison and Figure 11 in appendix for precision-recall. RPGAN and SN-ResNet perform with the same quality both in terms of FID and precision-recall curves.

| model | min FID | average FID |
|---|---|---|
| Five-bucket RPGAN | 16.9 | 20.8 |
| SN-ResNet | 16.75 | 18.7 |

Table 2: FID values for CIFAR-10.

To confirm that the layers of the standard GAN generator can be interpreted in the same way as the corresponding layers of its RPGAN counterpart, we perform the following experiment. We take a standard SN-ResNet GAN, consisting of five layers associated with the correspondent buckets in RPGAN, and train it on CIFAR-10. Then for each layer, we add normal noise to its weights. Intuitively, we expect that the noise injection in the particular layer would change generated samples in terms of characteristics, influenced by this layer. For instance, noise in the last two layers is expected to harm coloring scheme, while noise in the intermediate layers is expected to bring maximal semantic damage. Several samples, produced by perturbed layers, are presented on Figure 5. The images support the intuition described above and confirm that RPGAN may serve as an analysis tool for the corresponding generator model. Note, however, that injecting noise per se is not sufficient for interpretability. The perturbed generators produce poor images, which are difficult to analyze. Meanwhile, RPGAN always generates good-looking images, which allows to identify the factors of variation, corresponding to the particular layer. For instance, see Figure 8 for the colored MNIST dataset. Figure 8 (left) shows plausible images, generated by varying RPGAN blocks. In contrast, Figure 8 (right) demonstrates images from generators perturbed with small and large noise. For both noise magnitudes, these images are difficult to interpret. Of course, given the interpretations obtained via RPGAN, one can perceive similar patterns in the noisy generations, but noise injection alone is not sufficient for interpretability.

## 4.3 Incremental learning with RPGAN

In the next experiment, we demonstrate that the RPGAN model is also a natural fit for the generative incremental learning task (see e.g., Wu et al. (2018)). Let us assume that the whole train dataset $D$ is split into two disjoint subsets $D = D_1 \cup D_2$. Suppose that initially we have no samples from $D_2$ and train a generative model to approximate a distribution defined by the subset $D_1$. Then, given additional samples from $D_2$, we aim to solve an incremental learning task — to update the model with new data without re-training it from scratch. The RPGAN model allows solving this task naturally. First we train a generator with buckets $B_1, \ldots, B_n$ to reproduce the subset $D_1$. Once we want to extend the generator with samples from $D_2$, we add several new blocks to the buckets that are responsible for the features that capture the difference between $D_1$ and $D_2$. Then we optimize the generator to reproduce both $D_1$ and $D_2$ by training only the new blocks. Thus, instead of training a new generator from scratch, we exploit the pretrained blocks that are responsible for features, which

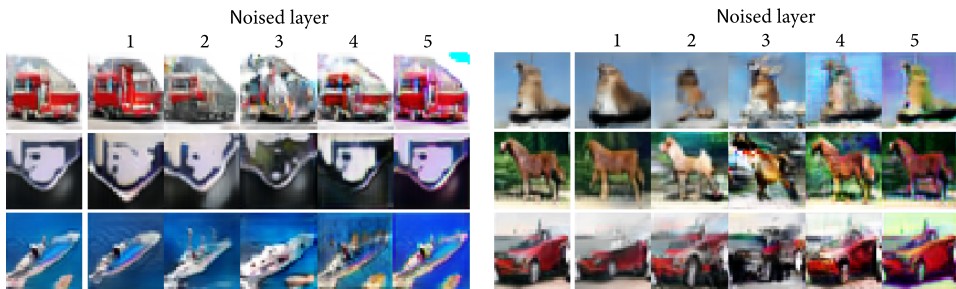

Figure 5: Images, produced by the standard SN-ResNet GAN with noise injection in the parameters of different generator layers. First column: original images, produced without generator perturbation. Other columns: images produced by generators perturbed in different layers.

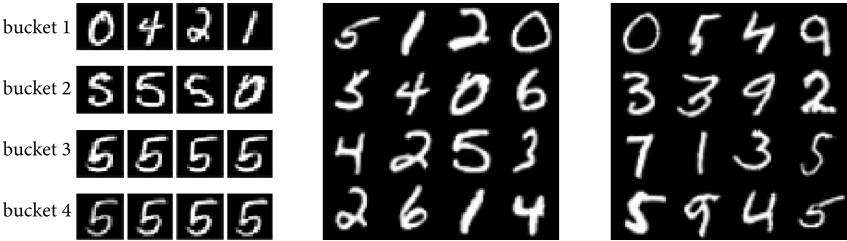

Figure 6: Incremental learning with RPGAN. *Left:* variations captured by different buckets with generator trained on $MNIST_{0-6}$. *Center:* samples picked from a generator trained on $MNIST_{0-6}$. *Right*: samples picked from a generator tuned on MNIST with only ten new blocks training (see details in the main text).

are common for $D_1$ and $D_2$. To illustrate this scenario, we take a partition of the MNIST handwritten digits dataset (LeCun, 1989) into two subsets $MNIST_{0-6}$ and $MNIST_{7-9}$ of digits from 0 to 6 and from 7 to 9 correspondingly. As for generator for $MNIST_{0-6}$ we take a 4-bucket RPGAN model with a number of blocks equal to 20, 20, 20, 8. Note that the last bucket is much thinner than the others, as it turns out to be responsible for variations in writing style, which does not change much across the dataset. Then we train the generator on the subset $MNIST_{0-6}$ of first 7 digits (see Figure 6, left and center). After that we add five additional blocks to each of the first two layers, obtaining a generator with a number of blocks equal to 25, 25, 20, 8 and pretrained weights in all blocks but five in the first and in the second buckets. Then we train the extended model to fit the whole MNIST by optimizing only the ten new blocks (see Figure 6, right).

## 4.4 PURELY LINEAR GENERATOR

As a surprising side effect of our model, we discovered that decent generation quality can be achieved by the RPGAN generator **with no nonlinearities**, i.e., one can train the RPGAN generator with all blocks consisting of linear transformations only. To demonstrate that, we take an RPGAN with the same ResNet-like generator architecture as in the experiments above. Then we replace all nonlinearities in the generator model by identity operations and train it on the CIFAR-10 dataset. The model demonstrates FID equal to 22.79 that is competitive to the state-of-the-art generative models of comparable sizes (see Figure 23 for generated images examples). Note that this approach fails for a standard GAN generator that maps a Gaussian distribution to an images distribution. Indeed, that generator would be a linear operator from a latent space to the images space with a Gaussian distribution in the images domain.

This purely linear generator architecture allows us to significantly speed up the image generation process for fully-connected layers. We group consequent buckets of fully-connected layers to form a new bucket. Blocks in the new bucket are linear transformations that are products of the blocks from the original buckets. To demonstrate this, we train a fully-connected generator network on the MNIST dataset (see Table 3 in appendix, left column). Then we join the last three buckets into a single one. We form a new bucket by blocks defined as the linear operators $B_{5k} \circ B_{4j} \circ B_{3i}$ where

$i, j, k$ are random indices of blocks from the buckets $B_3, B_4, B_5$ of the original generator. Thus instead of performing three multiplications of features vector from the second layer by matrices of the shapes $256 \times 512, 512 \times 1024, 1024 \times 784$ we perform a single multiplication by a $256 \times 784$ matrix. In our experiments, we achieved $\times 2.2$ speed up. Note, however, that after the compression, the latent space cardinality can decrease if a small subset of tuples $(i, j, k)$ is used to populate the new bucket. Nevertheless, as random products of joining buckets are used, we expect that the generated images would be uniformly distributed in the space of images, produced by the uncompressed generator (see Figure 24 for samples comparison).

## 5 CONCLUSION

In this paper, we address the interpretability of generative models. In particular, we have introduced RPGAN, an alternative design of generative adversarial networks, which allows natural interpretation of different generator layers via using random routing as a source of stochasticity. With experiments on several datasets, we provide evidence that different layers are responsible for the different factors of variation in generated images, which is consistent with findings from previous work. As a possible direction of future research, one can use the RPGAN analysis to construct efficient models, e.g., via identification of redundant parts of the generator for pruning or inference speedup.

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

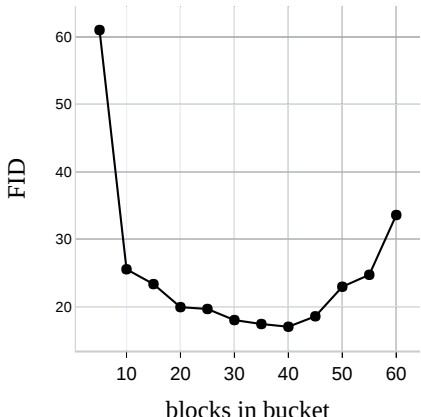

Figure 7: The FID values for RPGAN with different number of blocks.

Chenshen Wu, Luis Herranz, Xialei Liu, Joost van de Weijer, Bogdan Raducanu, et al. Memory replay gans: Learning to generate new categories without forgetting. In *Advances In Neural Information Processing Systems*, pp. 5962–5972, 2018.

Fisher Yu, Yinda Zhang, Shuran Song, Ari Seff, and Jianxiong Xiao. Lsun: Construction of a large-scale image dataset using deep learning with humans in the loop. *arXiv preprint arXiv:1506.03365*, 2015.

Matthew D Zeiler and Rob Fergus. Visualizing and understanding convolutional networks. In *European conference on computer vision*, pp. 818–833. Springer, 2014.

Jun-Yan Zhu, Taesung Park, Phillip Isola, and Alexei A Efros. Unpaired image-to-image translation using cycle-consistent adversarial networks. In *Proceedings of the IEEE international conference on computer vision*, pp. 2223–2232, 2017.

## A ADDITIONAL FIGURES AND EXPERIMENTS

### A.1 ABLATION ON NUMBER OF BLOCKS

Here we investigate the impact of the number of blocks in each RPGAN bucket on the generation quality. We train RPGAN with the SN-ResNet generator on CIFAR-10 with a different numbers of blocks in each bucket. For simplicity, all buckets have the same number of blocks. The resulting FID values are presented on Figure 7

If the number of blocks is too low, the resulting latent space appears to have insufficient cardinality to cover the dataset. On the other hand, a too high number of blocks results in a difficult training procedure and also fails.

### A.2 LSUN BEDROOM DATASET

To generate LSUN-bedroom-like images, we use the 7-bucket RPGAN with ResNet-like generator with five residual blocks, the first fully-connected layer, and the last convolutional layer. Similarly to CIFAR-10 experiments, during the generation, we freeze a random path and vary blocks in a single bucket to investigate its responsibility. See Figure 12 for blocks variations and Figure 13 for buckets responsibility analysis. Note that similarly to CIFAR-10, the central buckets have a maximal semantic impact. Last two buckets are mostly responsible for coloring. The first two buckets are responsible for local geometrical features. Note that here we face mode collapse for the third bucket: mainly, it affects only tiny local features. See Figure 14 for samples generated by the model.

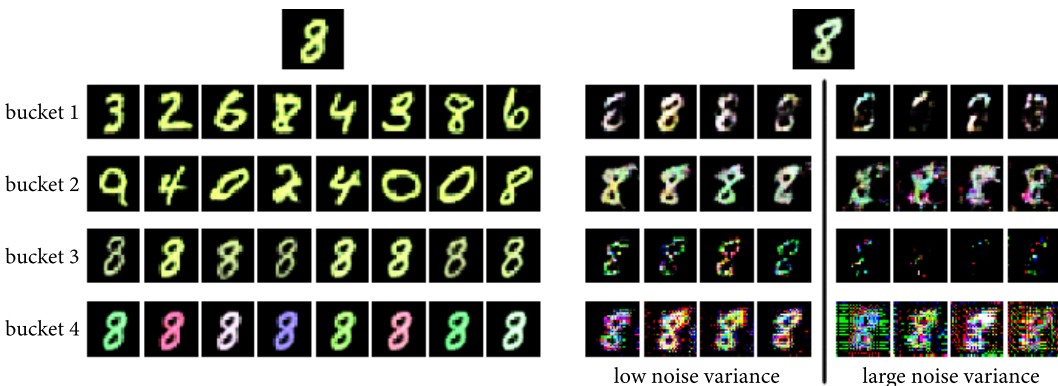

Figure 8: *Left*: images produced by varying blocks in a particular bucket of RPGAN. *Right*: images produced by the standard GAN after parameters perturbation in a particular generator layer, with low and high normal noise variance.

### A.3 ANIME FACES DATASET

Though this dataset is not standard for GANs, we use it in the experiments as it nicely reveals the RPGAN analysis tools. Here we use the 6-bucket RPGAN with ResNet-like generator with four residual blocks, the first fully-connected layer, and the last convolutional layer. See Figure 15 for block variations and Figure 16 for bucket responsibility analysis. Again, the content semantics is mostly defined by the intermediate buckets. The last two buckets are mostly responsible for coloring: the fifth bucket has the maximal impact on coloring, and the last bucket varies tones. The first buckets are responsible for small details (one can note the hair on the character's forehead). See Figure 17 for samples generated by the model.

### A.4 WASSERSTEIN GAN

Here we show that the concept of RPGAN works well with different generator architectures and learning strategies. Here we present plots for DCGAN-like generators consisted of consequent convolutional layers without skip connections. All the models were trained with the same parameters as described in Section 4. Despite of Spectral Normalization, we train these models as WGANs with weight penalty (Gulrajani et al., 2017). On the Figure 19 we show plots for a four-bucket generator trained on CIFAR-10. We also train four-bucket generator on colored MNIST, see Figure 8, left. Finaly, we show plots for the five-bucket generator and CelebA-64x64 dataset on Figure 21. See Figure 18, Figure 21, Figure 22 for buckets analysis.

## B RPGAN INTERPRETABILITY VS WEIGHTS NOISING

In this section we show that injecting noise in the generator weights cannot be used as a stand-alone interpretability method. Namely, we compare images produces by RPGAN and noise injection for models trained on randomly colored MNIST samples. We train both RPGAN and standard generators as a Wasserstein GAN with weights penalty.

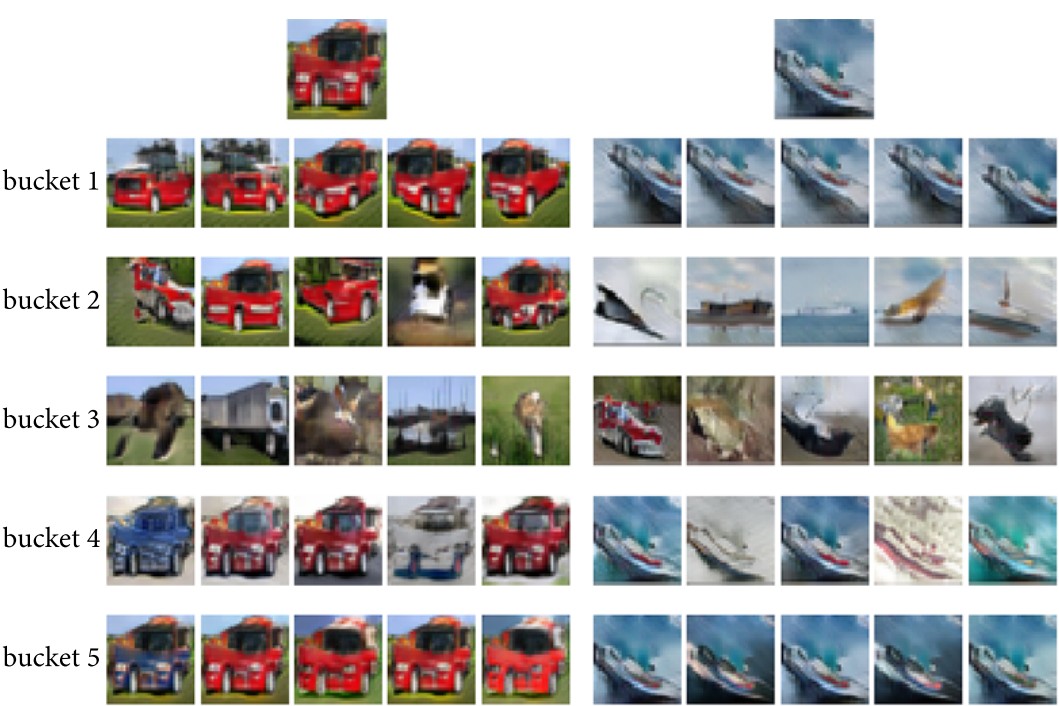

Figure 9: Frozen paths individual blocks variation in 5-bucket RPGAN.

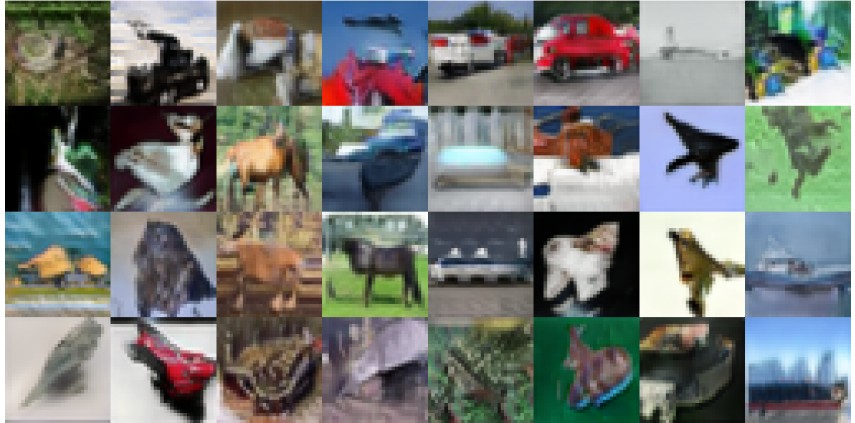

Figure 10: $32 \times 32$ samples generated by 5-bucket RPGAN.

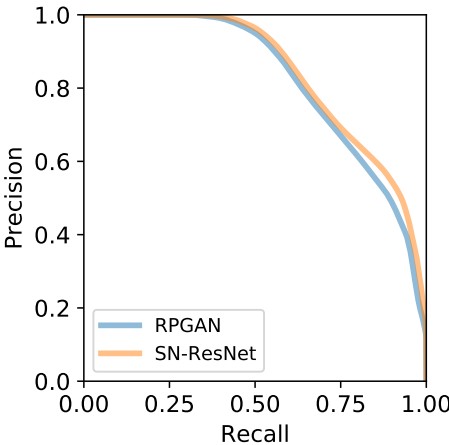

Figure 11: Comparison of precision-recall of RPGAN and its backbone SN-ResNet trained on CIFAR10.

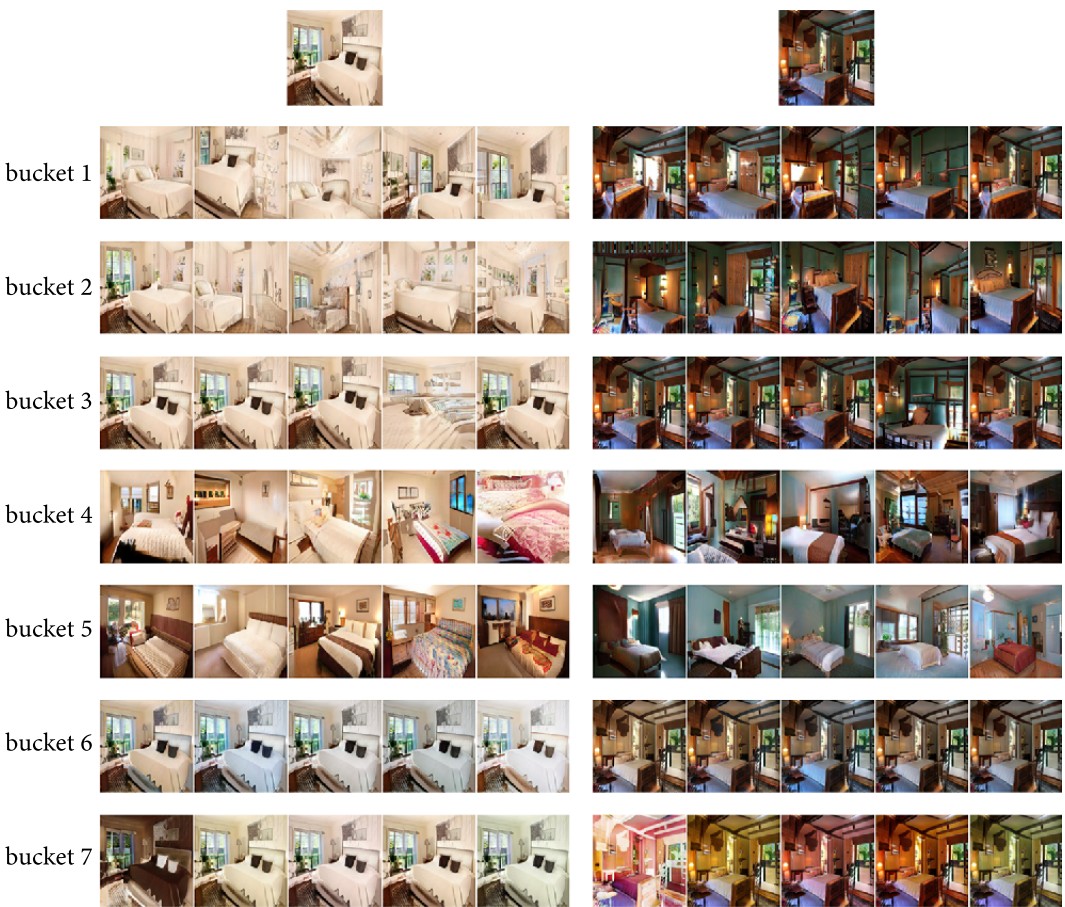

Figure 12: Frozen paths individual blocks variation in 7-bucket RPGAN.

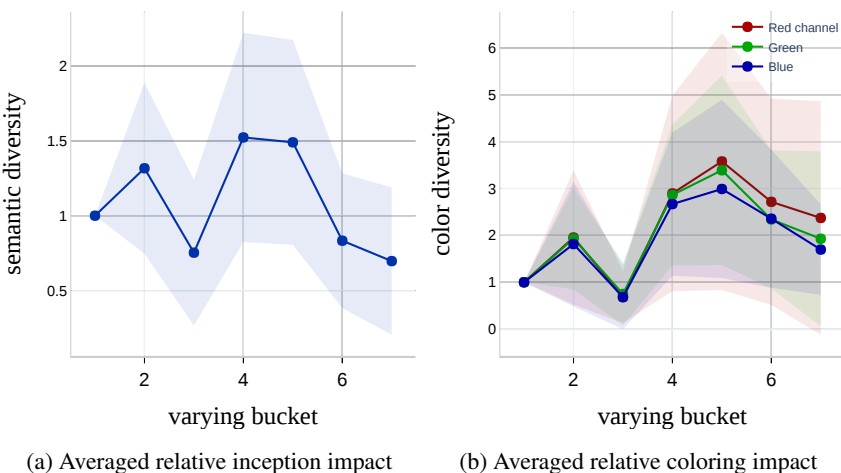

(a) Averaged relative inception impact      (b) Averaged relative coloring impact

Figure 13: LSUN bedroom buckets specification. See Section 4.1 for details.

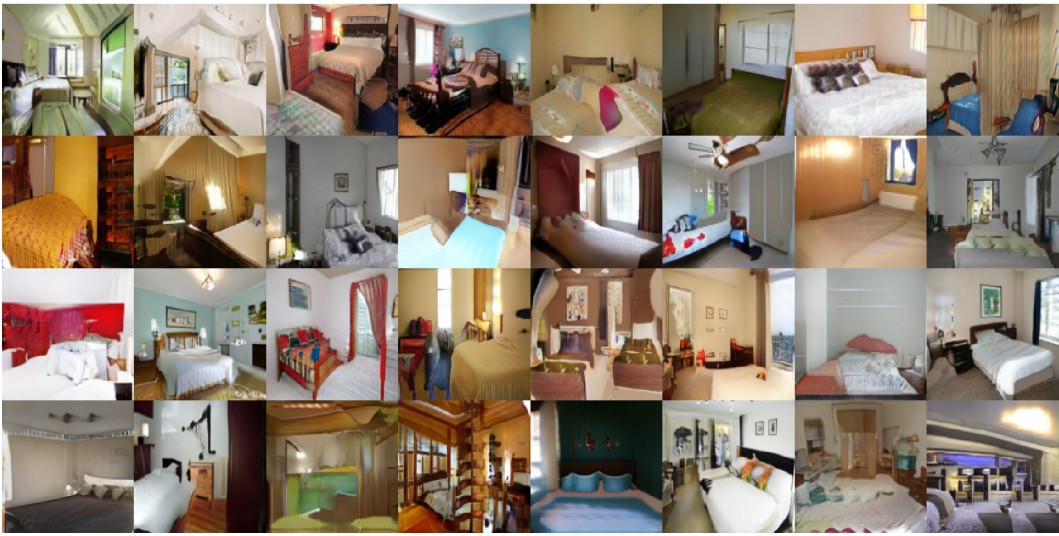

Figure 14: $128 \times 128$ samples generated by 7-bucket RPGAN

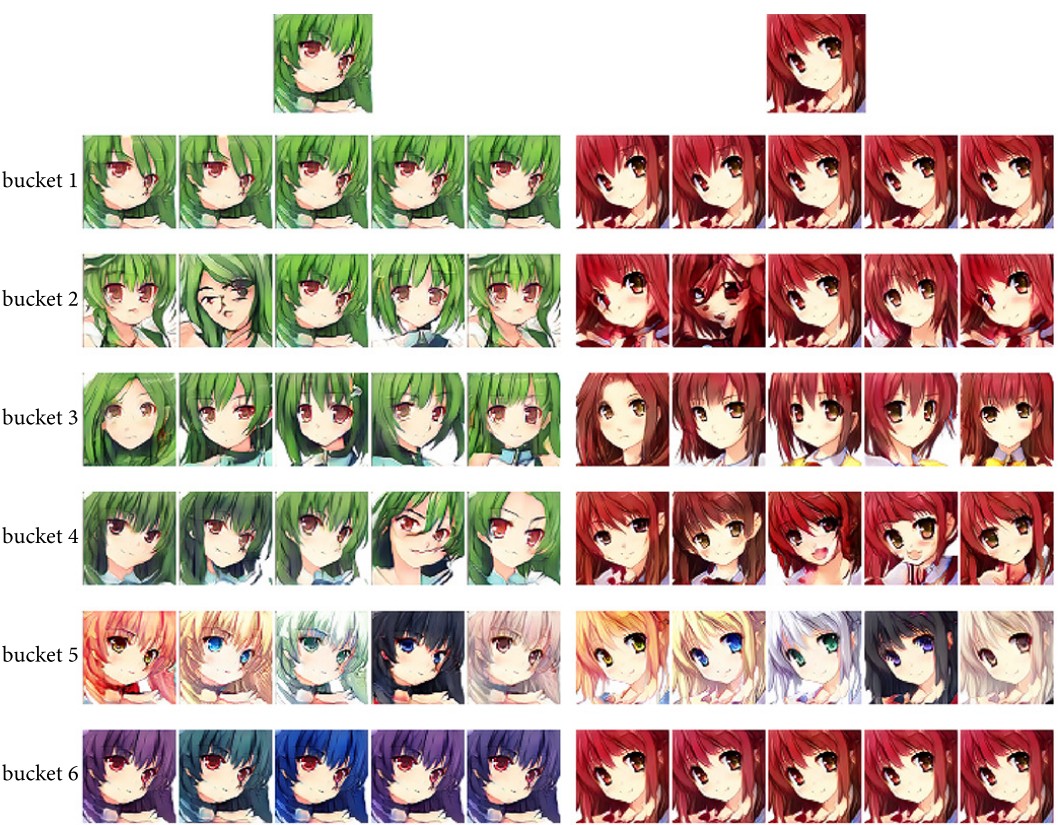

Figure 15: Frozen paths individual blocks variation in 6-bucket RPGAN.

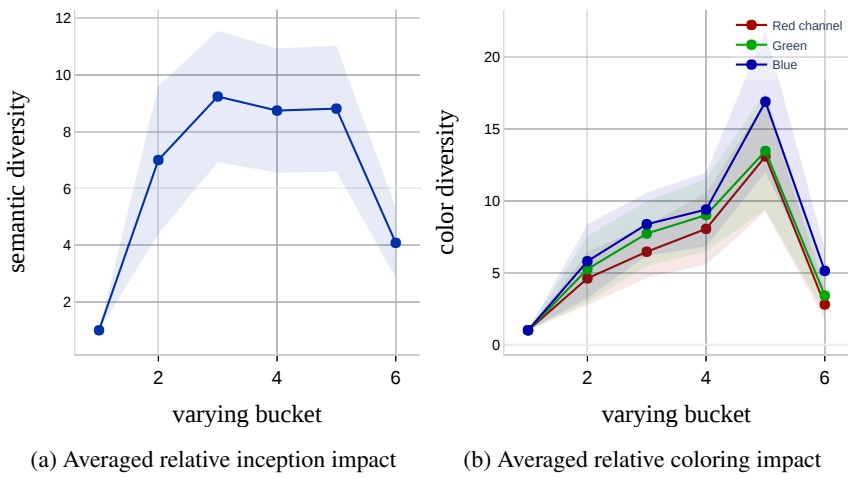

(a) Averaged relative inception impact

(b) Averaged relative coloring impact

Figure 16: Anime Faces buckets specification. See Section 4.1 for details.

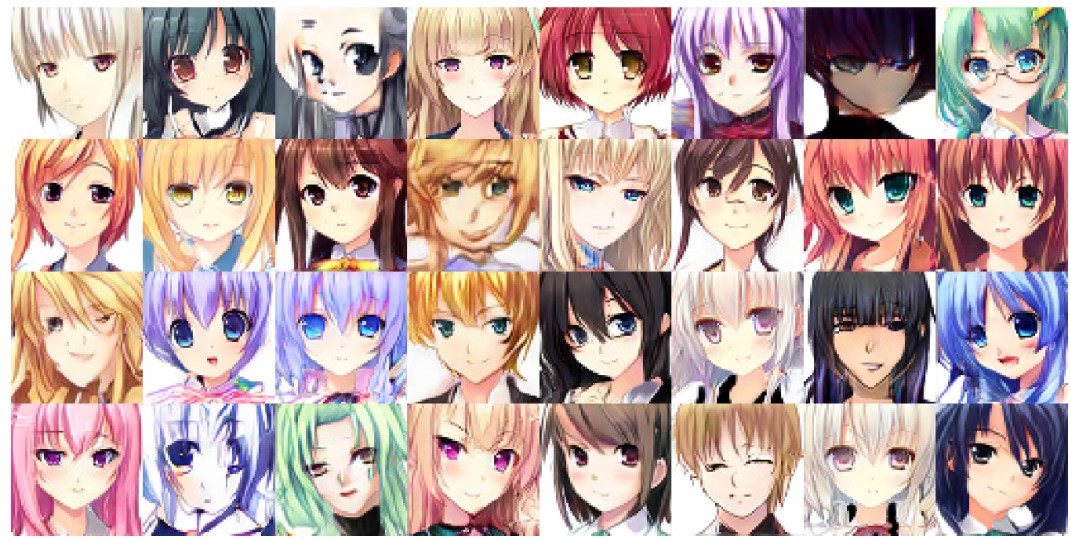

Figure 17: $64 \times 64$ samples generated by 6-bucket RPGAN.

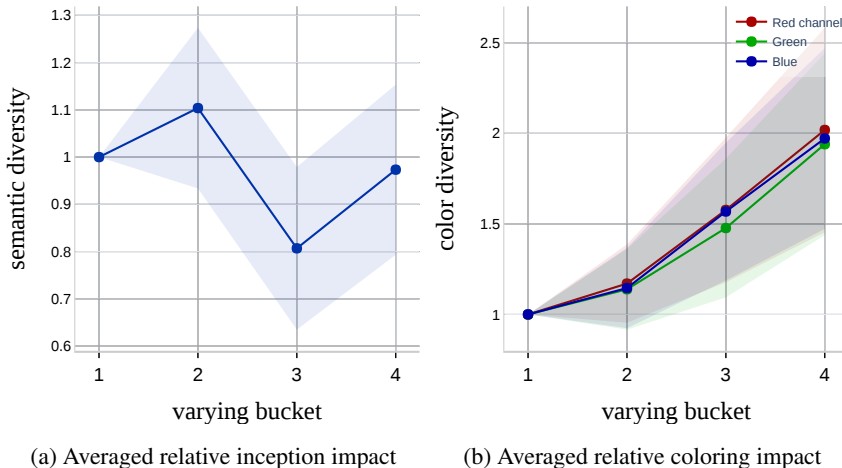

(a) Averaged relative inception impact

(b) Averaged relative coloring impact

Figure 18: Buckets specification of RPGAN with DCGAN backbone trained on colored MNIST. See Section 4.1 for details.

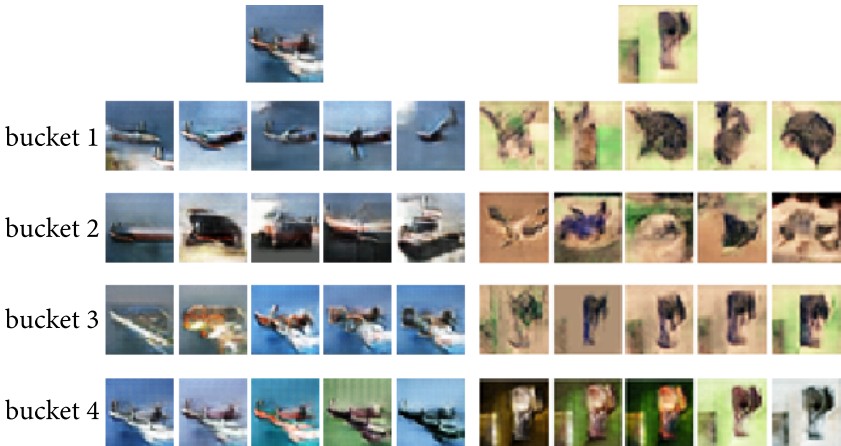

Figure 19: Frozen paths individual blocks variation in 4-bucket RPGAN with a DCGAN backbone. Generated images size is $32 \times 32$.

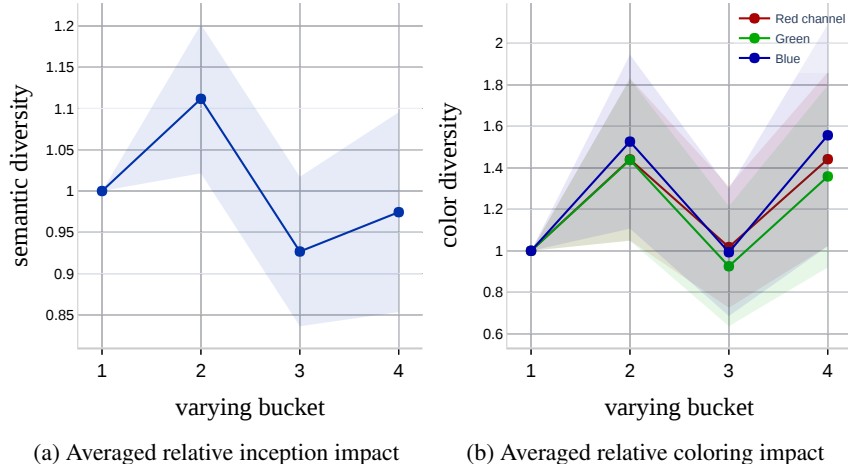

(a) Averaged relative inception impact

(b) Averaged relative coloring impact

Figure 20: Buckets specification of RPGAN with DCGAN backbone trained on CIFAR10 dataset. See Section 4.1 for details.

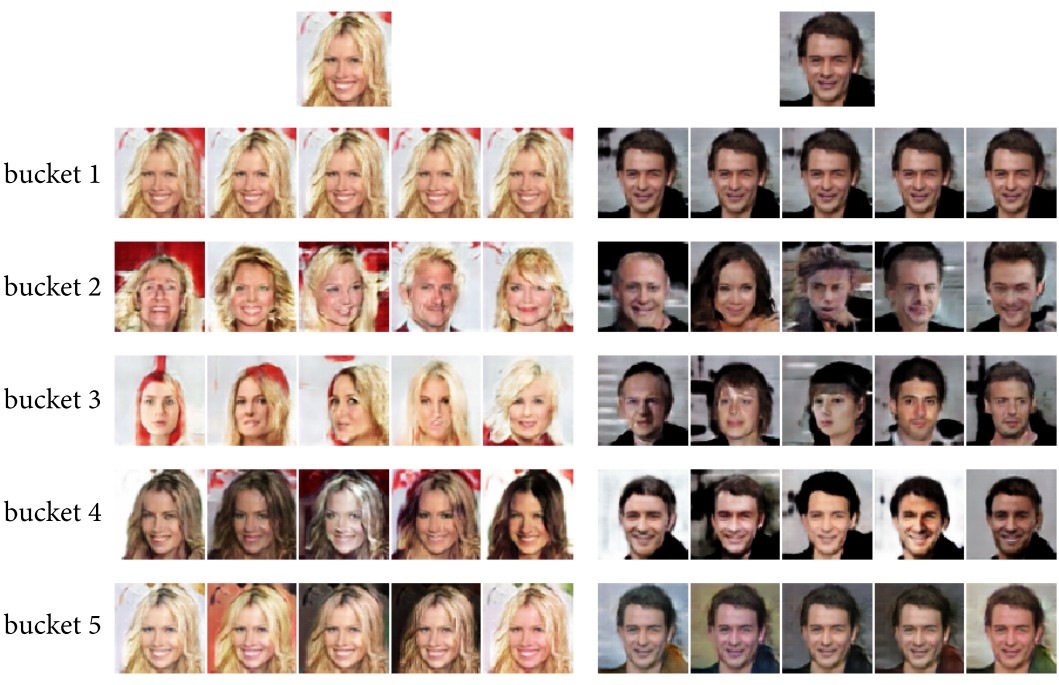

Figure 21: Frozen paths individual blocks variation in 5-bucket RPGAN with a DCGAN backbone. Generated images size is $64 \times 64$.

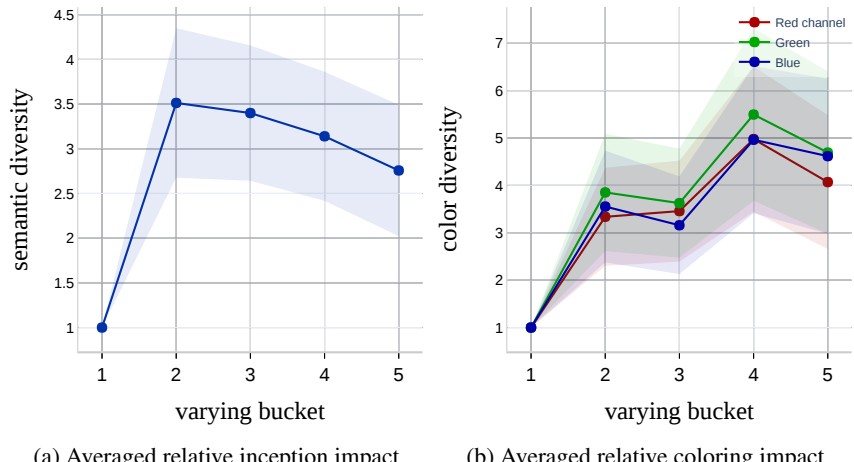

(a) Averaged relative inception impact  (b) Averaged relative coloring impact

Figure 22: Buckets specification of RPGAN with DCGAN backbone trained on CelebA dataset. See Section 4.1 for details.

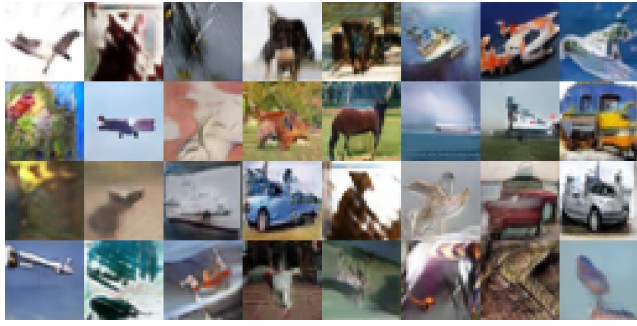

Figure 23: Samples from a 5-bucket ResNet-like generator without nonlinearities.

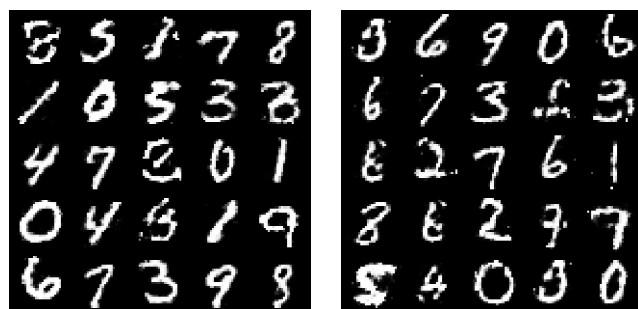

Figure 24: Digits generated by RPGAN without nonlinearities (*left*) and by its ×2.2 faster compression (*right*).

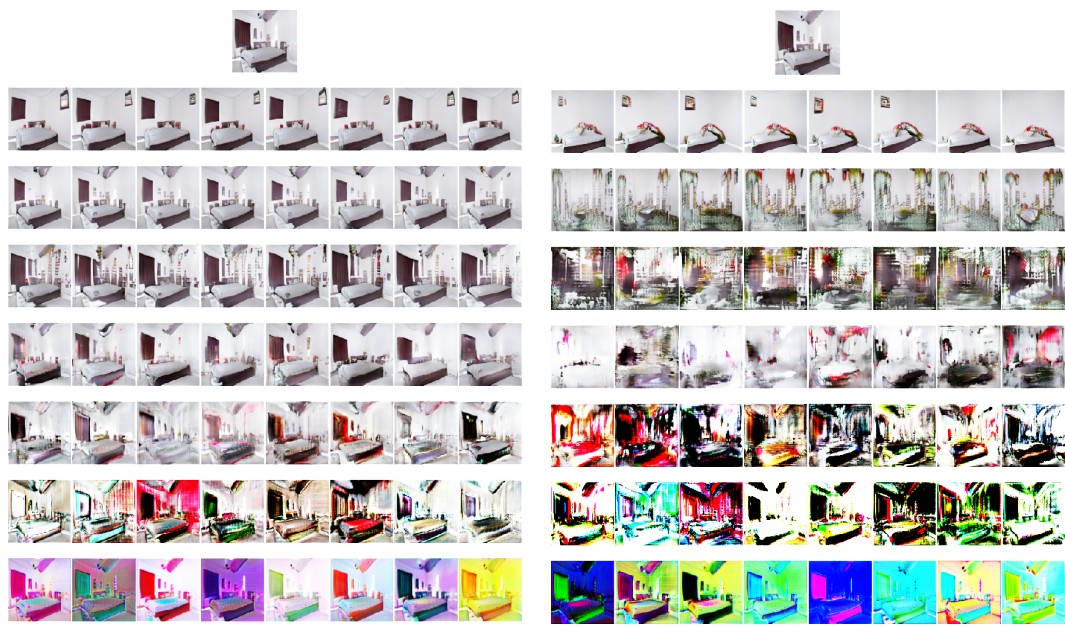

Figure 25: SN-ResNet generator with different layers noising. *Left*: low normal noise variance. *Right*: high normal noise variance

| Original | Compressed |
|---|---|
| $z \in \mathbb{R}^{128}$ ||
| fc, 32 blocks, 128 ||
| fc, 32 blocks, 256 ||
| fc, 32 blocks, 512 ||
| fc, 16 blocks, 1024 | fc, 128 blocks, 784 |
| fc, 16 blocks, 784 ||
| Tanh, reshape to $28 \times 28$ ||

Table 3: Fully connected RPGAN without nonlinearities and its compressed modification.

