# OpenReview forum: "RPGAN: random paths as a latent space for GAN interpretability"
_ICLR.cc/2020/Conference — Reject_

### Official Review · AnonReviewer1 · 2019-10-27
**Official Blind Review #1**

**Rating:** 3

**Review:**

This paper proposes the Random Path Generative Adversarial Network (RP-GAN) to serve as a tool for generative model analysis.  The main idea is to have several different buckets in each block of the generator and then train the generator with random paths. To interpret the features captured by each block, the authors unfreeze one block and show the variance of the generated images via different buckets.

The contribution of the paper is limited. Most of the observations proposed in this paper are just to confirm the findings in the recent paper Bau et al (2019). And the authors did not clarify why their methods are better than the previous work. In addition, this work changes some standard ways of generating images. For example,  they use a fixed input vector Z rather than a random vector Z following Gaussian distribution. Then when they claim that their findings are also valid to standard GAN generators, e.g., SN-ResNet, they add the noise to the weights in SN-ResNet and claim that we can conclude similar findings as RP-GAN. Therefore, my question is if adding noise to the weights to a generator is sufficient for the interpretation of the generator, why do we still need this work for further interpretation?

Minor:
In Figure 3, there are 10 images in each line but the number of buckets in each block is 40 for CIFAR10, as you claimed in Section 4. It should be clarified how you get those 10 images from 40 blocks in the unfreezed bucket.

In conclusion, I vote for a weak reject for this work.

**Experience Assessment:**

I do not know much about this area.

**Review Assessment: Checking Correctness Of Derivations And Theory:**

I carefully checked the derivations and theory.

**Review Assessment: Checking Correctness Of Experiments:**

I carefully checked the experiments.

**Review Assessment: Thoroughness In Paper Reading:**

I read the paper at least twice and used my best judgement in assessing the paper.

---

> ### Author Response · Authors · 2019-11-11
> **R1: We address your concerns.**
>
> Thank you for your time and comments; we address your concerns below.
>
> [Comparison to Dissection from Bau et al. (2019)]
> The proposed RPGAN tool is more general compared to the Dissection technique from Bau et al. (2019) since RPGAN can be directly applied to any dataset. In contrast, the Dissection requires a pretrained segmentation model. In particular, the authors use the model trained on the labeled ADE20K scene dataset to analyze GANs on the LSUN dataset. However, many standard benchmarks from the GAN literature lack appropriate labeled segmentation datasets, e.g., common CIFAR or MNIST datasets. Thus, one cannot directly use Dissection to interpret GANs on these datasets, while RPGAN provides interpretability in these cases, see Figures 6 and 8. We have added this explanation to a new revision.
>
> [work changes … standard ways to generate images]
> We consider the alternative source of stochasticity in RPGAN as a virtue:
>  - It does not degrade the model performance, see Table 2
>  - It provides interpretability of different parts of the generator network
>  - It can be applied for efficient incremental learning scenarios
>  - It allows to construct generators without non-linearities
> Overall, this is a model, and it is empirically shown to work well on several common datasets. To the best of our knowledge, our work is the first which demonstrates the advantages of alternative sources of generator stochasticity, and we expect that this idea per se will be interesting to the community.
>
> [if adding noise to the weights to a generator is sufficient for the interpretation of the generator, why do we still need this work for further interpretation?]
> We argue that injecting noise to the generator weights is *not* sufficient for the interpretability. The perturbed generators typically produce poor images, which are difficult to analyze. Meanwhile, RPGAN always generates good-looking images, which allows to identify the factors of variation, corresponding to the particular layer. For instance, see Figure 8 in a new revision. Figure 8 (left) demonstrates plausible images generated by varying RPGAN blocks. In contrast, Figure 8 (right) shows images produced by generators perturbed with small and large noise. For both noise magnitudes, these images are difficult to interpret.
> Of course, given the interpretations obtained with RPGAN, one can perceive similar patterns in the noisy generations, but only noise injection is not sufficient for interpretability. We have added a discussion in a new revision.
>
> [10 images instead of 40]
> We plot ten random images out of 40 to avoid excessively large figures. Have clarified in a new revision.

---

### Official Review · AnonReviewer2 · 2019-10-29
**Official Blind Review #2**

**Rating:** 3

**Review:**

This paper addresses the issue of interpretability of GAN generation through an alternative approach to the introduction of variability. To seed the generation, instead of providing a random input vector (typically sampled from a standard Gaussian distribution), the authors instead modify the generator architecture so as to allow for randomization in the routing: each layer is replaced by a bucket consisting of several blocks, and in forward propagation only through randomly chosen blocks. In this case, the input vector is chosen to be a constant - the only source of randomization
provided to the generator is in the choice of blocks through which to propagate. The explanability derives from the tendency of blocks to associate with a common interpretation after training. Their use of blocks necessitates the introduction of a block diversity loss, to discourage mode collapse. The scheme is referred to as RPGAN, for "Random Path GAN".

The main strengths of the paper:

(1) Their proposed approach is highly flexible. In principle, any underlying GAN architecture can be adapted by assigning each layer to a distinct bucket, and then replicating the layer across the blocks.

(2) Experimental results do show that different block sequences are associated with common image characteristics after training, especially for the initial and final layers.

(3) The use of non-standard ways of introducing stochasticity to GAN generation is an interesting idea in itself.

The main weaknesses:

(1) Although the authors provide experimental examples showing images associated with various paths through the architecture, is not clear how interpretations can be associated with these paths. In the examples presented, there seems to be a tendency for greater interpretability at the initial and final layers, with the explanation given for the intermediate layers being less convincing.

(2) The number of experimental examples is low, yet the authors draw rather firm conclusions (end of Section 4.1) regarding interpretability across layers. I am not sure that their conclusions adequately capture what is going on here, nor am I convinced that they generalize to other situations.

(3) The number of buckets limits the numbers of explanations. Essentially, the method has the same difficulties as in clustering, where specifying too many or too few groups can profoundly influence the nature and quality of the result. Although the authors do discuss an approach by which the number of buckets can be incrementally increased (thereby allowing for variation in the number of explanations generated), the experimental evidence is insufficient.

(4) Presumably, the replication of a layer across the blocks assigned to its bucket would require more training data and/or greater training times. What is the relationship in both time and quality between the original GAN network and its RPGAN versions?

(5) There are many presentational problems with this paper, in grammar, vocabulary and terminology, sentence structure, etc.

Overall, in its current state (not least due to presentational issues) the paper appears to be below the acceptance threshold.


**Experience Assessment:**

I have read many papers in this area.

**Review Assessment: Checking Correctness Of Derivations And Theory:**

I carefully checked the derivations and theory.

**Review Assessment: Checking Correctness Of Experiments:**

I carefully checked the experiments.

**Review Assessment: Thoroughness In Paper Reading:**

I read the paper thoroughly.

---

> ### Author Response · Authors · 2019-11-11
> **R2: We address your concerns.**
>
> Thank you for your time and comments; we address the items from your weaknesses list below.
>
> 1) [is not clear how interpretations can be associated with these paths.]
> To avoid possible confusion: the interpretations are associated with buckets, not with paths. Varying active blocks in each bucket we get an understanding of what factors of variations are captured by this bucket. For instance, in Figure 3, using different blocks in the fourth bucket results in images with the same content but of different colors, which indicates that the fourth bucket is “responsible for” coloring. As another example, varying blocks in the first bucket shows that this bucket mostly determines an object location on the image.
>
> (2) [I am not convinced that the conclusions generalize to other situations.]
> We do not claim that the layers’ roles identified in the section 4.1 are the same for all datasets and generator architectures. Moreover, Figure 6 (left) of the original submission demonstrates that on MNIST, the first bucket is responsible for image semantics, while on CIFAR10 semantics is determined by the intermediate buckets.
> However, we argue that the RPGAN tool is general: it allows to analyze the roles of different generator layers on any datasets. As an addition, we accomplish the RPGAN results with a number of experiments on other domains, model architectures and learning strategies. Namely, we show that the concept works for DCGAN-like generators trained as Wasserstein-GP GANs on colored MNIST, CIFAR10 and CelebA datasets (see Figures 8, 19, 21).
>
> (3) [The number of buckets limits the numbers of explanations.]
> To avoid confusion between \textit{buckets} and \textit{blocks}: each bucket corresponds to the particular generator layer, while different blocks denote different replications of this layer inside the bucket. The number of buckets hence equals to the number of layers in the generator. The number of blocks in all our experiments equals 10-40, e.g., see Figure 7. In all our experiments, a few dozens of images are enough to understand the factors of variations, corresponding to the particular buckets.
>
> (4)[What is the relationship in both time and quality between the original GAN network and its RPGAN versions?]
> In all our experiments, we use completely the same training protocols for both RPGAN and the original GAN. Both models are always trained on the same data, with the same number of steps for generator/discriminator, etc, see the beginning of Section 4. In terms of wall-clock time, there were no differences in training time for both models on our hardware and the implementations from the code https://github.com/rpgan-ICLR2020/RPGAN.
>
> (5) In a new revision, we have largely rewritten the text and expect it to be easier to follow.

---

### Official Review · AnonReviewer3 · 2019-10-29
**Official Blind Review #3**

**Rating:** 8

**Review:**

This paper presents a variation on the generator of GANs. The authors modify the generator by adding a concept of "blocks" which are randomly activated based on part of the random input vector. It is similar to adding random dropout in the generator, except that the dropout would apply to larger sets of activations instead of single component.

The authors also add a diversity term to force the different blocks to have different blocks. This is a term based on the L2 distance between the weights of different blocks. Although this term is ad-hoc and could probably be refined into something more grounded in theory, it should indeed provide some diversity.

This block structure allows for more understanding of what each layer of the generator does, since it is easy to change the discrete variables that switch blocks. The paper presents an empirical evaluation of what each switch does, and show that concepts are well disentangled between layers (for instance, one layer changes the background whereas another one changes the color of the foreground).

They also show that blocks can be added after training is done which is a nice property for incremental training.

They also show that this framework can train a generator without non-linearities (except for the block switching), which could potentially simplify the analysis of such networks.

Generated samples are presented up to 128x128 pixels which, although far from state of the art, proves that the concept works.

**Experience Assessment:**

I have published in this field for several years.

**Review Assessment: Checking Correctness Of Derivations And Theory:**

I assessed the sensibility of the derivations and theory.

**Review Assessment: Checking Correctness Of Experiments:**

I assessed the sensibility of the experiments.

**Review Assessment: Thoroughness In Paper Reading:**

I read the paper at least twice and used my best judgement in assessing the paper.

---

### Author Response · Authors · 2019-11-11
**Common answer to the reviewers.**

We thank the reviewers for their comments. We have uploaded a new version and we summarize the major changes below. We also address the individual concerns in separate comments.

1. We describe the advantages of RPGAN compared to Bau et al. (2019) in related work.
2. We explain and experimentally confirm that the noise injection cannot serve as a stand-alone tool for interpretability in section 4.2.
3. We add the experimental results on more datasets and generator architectures, demonstrating wide applicability of RPGAN.
4. Several sections were rewritten and now we expect the submission to be easier to follow.

---

### Decision · Program_Chairs · 2019-12-19

**Decision:**

Reject

**Comment:**

The paper received mixed scores: Weak Reject (R1 and R2) and Accept (R3). AC has closely read the reviews/comments/rebuttal and examined the paper. After the rebuttal, R2's concerns still remain. AC sides with R2 and feels that the generated interpretations are not convincing, and that the conclusions drawn are not fully supported. Thus the paper just falls below the acceptance threshold, unfortunately. The work has merits however and the authors should revise their paper to incorporate the constructive feedback.